# Insights into the Secretome of Mesenchymal Stem Cells and Its Potential Applications

**DOI:** 10.3390/ijms20184597

**Published:** 2019-09-17

**Authors:** Sharon Eleuteri, Alessandra Fierabracci

**Affiliations:** Infectivology and Clinical Trials Area, Children’s Hospital Bambino Gesù, Viale San Paolo 15, 00146 Rome, Italy; sharon.eleuteri@gmail.com

**Keywords:** mesenchymal stem cells (MSCs), extracellular vesicles (EVs), secretome, immunomodulatory properties, regenerative properties, miRNA, therapy, safety, clinical trial

## Abstract

Mesenchymal stem cells (MSCs) have regenerative, immunoregulatory properties and can be easily isolated and expanded in vitro. Despite being a powerful tool for clinical applications, they present limitations in terms of delivery, safety, and variability of therapeutic response. Interestingly, the MSC secretome composed by cytokines, chemokines, growth factors, proteins, and extracellular vesicles, could represent a valid alternative to their use. It is noteworthy that MSC-derived extracellular vesicles (MSC-EVs) have the same effect and could be advantageous compared to the parental cells because of their specific miRNAs load. MiRNAs could be useful both in diagnostic procedures such as “liquid biopsy” to identify early pathologies and in the therapeutic field. Not only are MSC-EVs’ preservation, transfer, and production easier, but their administration is also safer, hence some clinical trials are ongoing. However, much effort is required to improve the characterization of EVs to avoid artifacts and guarantee reproducibility of the studies.

## 1. Introduction

MSCs are multipotent, self-renewable, and able to differentiate into osteoblasts, chondrocytes, and adipocytes (tri-lineage differentiation) [1]. MSCs are found in bone marrow, muscle, and adipose tissue [2], and also in Wharton’s jelly [3], dental pulp [4], peripheral blood [5], skin [6], lungs [7], chorionic villi [8], menstrual blood [9], and breast milk [10]. These human, plastic-adherent and spindle-shaped cells can be detected and isolated through the expression of specific surface antigens (CD (cluster of differentiation) 105+, CD73+, CD90+, CD45-, CD34-, CD14-/CD11b-, CD79α-/CD19-, HLA-DR-) as established by the Mesenchymal and Tissue Stem Cell Committee of the International Society for Cellular Therapy (ISCT) [11]. Despite having a similar phenotype, MSCs from various tissues show additional identifying features that reflect their different parental sources. More studies are required to exhaustively unravel their interesting properties, thus nowadays they represent the most investigated population of adult stem cells [2]. Moreover, MSCs are easy to isolate [12] and expand in vitro [2], making them an attractive tool for clinical applications.

Due to their ability to restore tissue damage, promote regeneration and tissue homeostasis [13], in recent years MSCs have become pivotal for therapies focused on heart damage repair [14]. Additional studies, first in vitro and then in vivo, on animal models of autoimmune diseases have proved that MSCs are able to interfere with the proliferation, activation, and function of immune cells by altering both innate and adaptive immunity mechanisms [15]. However, a large number of clinical trials have reported that, despite a functional improvement of the target tissue after MSC transplantation, engraftment and proper differentiation of MSCs may not occur [2]. In this regard, efficient delivery of MSCs, i.e., for bone regeneration, requires a suitable scaffold such as a matrix [16].

The evident therapeutic efficacy of MSCs does not seem to rely on the physical proximity of the transplanted cells within tissues [17]. Many studies have confirmed that the synergistic action of small molecules secreted by MSCs is able to reduce cell injury and improves tissue repairing capacity [18,19]. Close proximity of the mother cells is not required since their soluble molecules are vehicled to the target. Therefore, MSC secretome, including MSC-derived extracellular vesicles (EVs) (MSC-EVs), attracted more attention than the MSC transplantation itself. EVs represent an intercellular communication system, used by many cell types, which are able to carry many molecules in a single package to a specific target cell, thus themselves can also be useful in therapy [1]. Medical applications of the vesicles change with the composition and structure of the mother cell [20]. In this regard, MSC-EVs represent a powerful tool that is able to maintain or restore tissue homeostasis [2] and directly interact with cells of the immune system regulating their activity [1]. Thanks to the advantages of their use compared to the parental stem cells, many studies have highlighted several potential applications of MSC-EVs.

In this review we focus on the relevance of the MSC secretome with special reference to miRNAs and their potential application as an alternative to MSCs.

## 2. Limits of Mesenchymal Stem Cell-Based Therapies

Lately, MSC injections-based therapies are considered very promising; thus, thousands of clinical trials are underway (http://www.clinicaltrials.gov/). However, so far, many of these studies have not reached the required endpoints. In this regard, using MSCs to prevent cardiac damage or to allow cardiac repair, have led to insufficient cardiomyocyte or vascular cell differentiation, despite MSC beneficial effects being clearly demonstrated [21]. Clinical trials have also concerned kidney diseases [22], the spinal cord [23], liver injury [24,25] and inflammatory diseases, such as Graft-versus-Host-Disease (GvHD) [26,27], Crohn’s disease [28,29], rheumatoid arthritis [30] and lupus nephritis [31].Thus, despite the enthusiasm for MSC potential in tissue repair, currently there are several limitations in MSC applicabilitydue to conflicting results in clinical settings.

Remarkably, the issue of safety in delivering MSCs has not been fully unraveled. Some researchers have demonstrated that the administration of MSCs intravascularly is feasible [32,33,34], but other studies have revealed that engrafted cells can maldifferentiate and generate tumors [35,36]. Thirabanjasak et al. [37] reported a case of lupus nephritis treated with autologous stem cells that developed stem-cell-derived or –induced angiomyeloproliferative lesions.

MSC therapeutic action is influenced by hypoxic preconditioning, 3D cell culture scaffold or addition of stimuli that modify the production of different factors by these cells [38,39]. Indeed, the secretome is influenced by the micro-environment characteristics that MSCs can encounter within a human or animal organism, thus conditioning the therapeutic response. In this regard, Shi et al. [13] suggested evaluating the inflammatory status of patients recruited for clinical trials—the response of MSCs indeed depends on the stage (early, latent, end-stage) of disease that may present with a different inflammatory microenvironment.

Another difficulty is the possible entrapment of MSCs or their byproducts in microvasculature. Some studies in both humans and animals have reported that MSCs were stuck in lung arterioles and MSC debris was detected in capillaries. Nevertheless, in humans the consequent risk of thrombosis was lower than in rats [40].

## 3. Molecules and Enzymes of MSCs Secretome

### 3.1. Molecules of MSC Secretome

In the light of the foregoing, the paracrine hypothesishas become more consolidated, leading to the investigation of the secretome of MSCs and their applicability in regeneration and immunomodulation.

The secretome of MSCs includes cytokines, chemokines, growth factors, anti-inflammatory factors, and even proteins conveyed by EVs [15]. Therefore, the search for the molecules responsible for its activity is of upmost importance for its possible applications (Table 1). Secreted factors can produce different effects, and some are released only in certain micro-environments.

Cytokines responsible for immunomodulation include TGF-β (tumor growth factor β) and mpCCL2 (MMP (metalloproteinase)-processed CCL2 (C–C motif chemokine ligand2)).

TGF-β has a key role in the activation of regulatory T cells (Tregs) [41]; indeed, Tregs regulation is overruled by adding an anti-TGF-β antibody or TGF-β siRNA in MSC co-cultures [42]. In the same study [42], TGF-β was shown to stimulate Tregs both in vitro and in vivo in SLE patients. In addition, TGF-β is involved in inhibiting the immune response by suppressing dendritic cell (DC) maturation [43] and T-helper 17 (Th17) cell generation [44]. MSCs secrete metalloproteinases (MMPs) that are able to halt CC (C–C motif) chemokines such as CCL2 (targeting chemokine (C–C motif) ligand 2). After MMP-mediated cleavage, CCL2 does not work as an agonist, but as an antagonist of T cell chemotaxis and activation [45].

Recently, MSCs were shown to secrete IL-1Ra (interleukin 1 receptor antagonist), a competitive inhibitor of IL-1 (Interleukin-1). Its production induces M2-like macrophage polarization, inhibits B cell differentiation [46] and stalls TNF-α production by macrophages in vitro [47].Growth factors including VEGF (vascular endothelial growth factor), HGF-1 (Hepatocyte growth factor-1),and LIF (leukemia inhibitory factor) have also been identified within the secretome, which have regenerative potential.

VEGF is involved in functional recovery after cerebral ischemia by reducing infarct size and increasing endogenous neurogenesis and angiogenesis at the same time [48]. The activation of receptor tyrosine kinase Met through HGF-1 enhances angiogenesis, myogenesis, and hematopoiesis [49]. Li et al. [50] demonstrated the renotropic and protective function of HGF in a kidney injury animal model. Furthermore, HGF-1 with LIF interfere with differentiation of T helper cells [13]. In particular, LIF is considered a potential therapeutic target for multiple sclerosis: Th17 cells are involved in the disease pathogenesis and LIF, through the SOCS3–STAT3(suppressor of cytokine signaling 3—signal transducer and activator of transcription 3) pathway, can inhibit STAT3 phosphorylation and, lastly, Th17 cell differentiation [51]. Additionally, LIF restores lymphocyte and Foxp3 (Forkhead box P3)+Tregs proliferation [52].

Furthermore, MSC secretome includes PGE2 (prostaglandin E2) which is known to exert multiple functions. It induces macrophages to produce anti-inflammatory IL-10 (Interleukin-10) [53], which, in turn, suppresses NK (Natural Killer) [15] and T helper cell [54] proliferation. Interestingly, PGE2 affects macrophage metabolic status and plasticity [55]. Recently, PGE2 secreted by MSCs was shown to promote hepatocyte proliferation and reduce apoptosis in a mouse model of acute liver failure (ALF) [56]. In other studies, TSG6 (tumor necrosis factor-inducible gene 6 protein) was critical for tissue repair in peritonitis [57] and experimental autoimmune encephalomyelitis (EAE) [58]. It is noteworthy that TSG6 not only interferes with the migration of leucocytes, but also induces repair of corneal epithelial cell damage in a diabetic corneal epithelial wound [59].

In literature, the galectin network was shown to mediate the immunomodulatory effects of MSCs [60,61]. In particular, Gal-1 (galectin-1) is constitutively expressed in MSCs and counteracts the proliferation of CD4+ and CD8+ T cells [60]. Meanwhile, Gal-9 (galectin-9) is induced by proinflammatory cytokines and leads to T cell apoptosis by binding glycoreceptor TIM-3 (T cell immunoglobulin domain and mucin domain-3) [61]. Gal-1 and Gal-9 are expressed both intracellularly and in MSC culture supernatants [60].

### 3.2. Enzyme Activity of MSCs Secretome

MSC immunoregulation also involves enzyme activities (Table 2). IDO (indoleamine 2,3-dioxygenase) and NOS (nitric oxide synthase) mediate immunosuppression in the human and murine organism, respectively. IDO is a metabolic enzyme that converts tryptophan into toxic kynurenines and leads to the depletion of this amino acid which is essential for T cells [62]. NOS produces NO (nitric oxide) that is able to stop the cell cycle through the JAK-STAT (Janus kinase/signal transducers and activators of transcription) pathway; additionally it modulates the activity of MAPK (mitogen-activated protein kinase) and NF-κB (nuclear factor kappa-light-chain-enhancer of activated B cells), thus leading to cell apoptosis [13,63].

HO-1 (heme oxygenase-1) is another enzyme whose metabolic products have distinct and specific properties [64,65]. In particular, it degrades heme into biliverdin, CO (carbon monoxide), and iron ions [65]. Biliverdin compromises infiltration and T cell proliferation [66] while CO prevents the expression of inflammatory cytokines in macrophages [65]. Chabbanes et al. [64] showed that HO-1 mediates the inhibition of T-cell proliferation and prolongs survival in a rat model of cardiac allotransplantation. Zhang et al [67] also demonstrated a protective effect of HO-1 on acute liver failure.

## 4. From MSCs to EVs: Toward Cell-Free Therapies

The immunosuppressive effects of MSCs on specific immunotypes (including T, B lymphocytes and NK cells) can be reproduced by extracellular vesicles isolated from MSC culture supernatants, providing a valid alternative to direct MSC use [62,68,69]. Even if EVs use different action mechanisms compared to their parental cells, the resulting effects are either similar or even enhanced. Indeed, EVs contain MSC secreted products, but also specific molecules protected by their phospholipid bilayer [1,15,62,68].

The potential advantageous use of EVs could also be exploited in regenerative therapies based on preliminary experimental observations. Nakamura et al. [70] demonstrated that MSC-EVs are responsible for myogenesis and angiogenesis processes in a cardiotoxin muscle injury model; this effect is attributed to the miRNA content of vesicles, instead of paracrine MSC molecules. 

Systemic administrations of MSC-EVs induced neurogenesis, angiogenesis, and functional recovery in a mouse model of traumatic brain injury [71]. MSC-EVs administration produced similar effects in the treatment of cerebral ischemia in C57BL6 mice compared to MSC-based therapy, inducing long-term neuroprotection [72].

The application of MSC-EVs in therapy can be more advantageous than MSCs alone for several reasons: They cannot proliferate, are simple to preserve and transfer, and are of easier production [73]. In addition, EV intravenous administration appears to be safer [74]; EVs are not toxic [75] with low immunogenicity [76]. Some clinical trials are ongoing for the treatment of acute ischemic stroke (NTC03384433), Type 1 diabetes (NTC02138331), and refractory macular holes (NTC03437759).

However, avoiding artifacts and guaranteeing the reproducibility of the studies are challenging since no techniques are available to ensure the absolute purification and characterization of EVs. First of all, EV experts have proposed in 2018 MISEV (Minimal Information for Studies of Extracellular Vesicles) guidelines [77] the most appropriate methods to separate EVs from non-EV components based on biofluids, level of recovery, and specificity. To date, ultrafiltration and size-exclusion chromatography [78,79], ultracentrifugation [80,81], and immunoaffinity are widely used methods [82]. Extensive studies are necessary to improve specific technologies. For example, Chen et al. [83] proposed a dielectrophoretic chip-based method to isolate EVs for diagnostic purposes. 

Furthermore, MISEV guidelines summarize the recommendations on how to carry out proper characterization of the vesicles used in pre-clinical and clinical investigations. This relies on the presence of at least one protein within each of the following categories:1)transmembrane or GPI-anchored proteins e.g., tetraspanins or integrins;2)cell specific proteins e.g., CD45 (immune cell), TSPAN8 (epithelial cell), ERBB2 (breast cancer);3)cytosolic proteins in EVs e.g., ESCRT I/II/III or ALIX.

Analysis of co-isolated contaminants such as lipoproteins in plasma or serum, Tamm-Horsfall protein in urine, is additionally required to assess sample purity.

## 5. A Look at Extracellular Vesicles (EVs)

EVs represent a sophisticated communication system for cells: Thanks to their ability to carry key molecules, they affect the physiological and pathological functions of recipient cells [69].

EVs are classified according to their origin in microvesicles (MVs) and exosomes [15]. MVs formation takes place through direct outward budding associated with an asymmetric calcium-induced reorganization of the phospholipids inside the plasma membrane [84]. In particular, phosphatidylserine is moved towards the outer layer to induce localized curvatures [20]. In addition, cytosolic proteases such as calpain and gelsolin, get activated and then cut the protein network of the actin cytoskeleton, allowing membrane budding [85]. Conversely, exosomes are formed intracellularly, starting from endosomes that merge with endocytic vesicles, thus incorporating their content. Undergoing a series of alterations, first endosomes become late endosomes, also called multivesicular bodies (MVBs), which are distinguished by small interluminal vesicles (ILVs). After MVB maturation, ILVs merge with the plasma membrane and become exosomes released in the extracellular space [20].

The different origin of exosomes and MVs determine their dissimilar size (exosomes: 40–120 nm; MVs: 50–1000nm) and content [86]. Microvesicles are more heterogeneous than exosomes; they contain phosphatidylserine, metalloproteinases, some integrins and selectins (P-selectin) [87]. Instead, exosomes contain proteins with GTPase activity involved in transportation and fusion (Annexin, Rab proteins) [73], heat shock proteins (Hsp 60, Hsp70, Hsp90), and proteins co-responsible for the vesicle biogenesis (Alix, TSG101) [20,73,75]. Exosomes also express tetraspanins (CD63, CD81, CD9) required in the fusion between exosomes and recipient cells [88], the main component of TEMs (tetraspanin-enriched microdomains). The microdomains create associations with various proteins on the cell surface, regulating access to functional regions of the cell membrane [89].

These proteins were identified by means of several techniques including Western blotting to identify tetraspanins CD9, CD63, and CD81 [90] and mass spectrometry [91]. Conventional flow cytometry with beads to capture EV sand detect surface markers by fluorescent antibodies represents a valid alternative [92].

In addition, a complete characterization of single vesicles focusing on morphology or specific properties is possible by combining imaging techniques i.e.,transmission electron microscopy [93] or atomic force microscopy [94]; light scattering [95] gives information about biophysical parameters.

The proteomic characterization of MSC-EVs have confirmed the presence of the proteins mentioned above, i.e., specific markers of MSCs, but also proteins that are not expressed in parental cells [96,97,98]. Using the PANTHER software, Mardpour et al. [73] grouped 938 proteins, annotated in the Exocarta database, into 11 categories. Among those categories, the metabolic and cellular processes show the highest scores. Furthermore, 44 proteins appear to interact with the immune system and others belonging to the PI3K (phosphatidylinositol 3-kinase)/Akt (protein kinase B)pathway (RTK(receptor tyrosine kinase), Grb2(growth factor receptor bound protein 2), Ras, ITGA (integrine alpha), ITGB (integrine beta), Rac1 (Ras-related C3 botulinum toxin substrate 1), CPRC (Cysteine proteinase 3 precursor), and Gβγ). This pathway is directly related to growth, survival, metabolism, and defense mechanisms of the innate immune system [99].

Interestingly, a significant number of 20Sproteasome members was found in EVs [73]. These proteins are transported inside the recipient cell to form a multiprotein complex [62]. This degrades misfolded proteins or pathogenic peptide aggregates co-responsible for, e.g., cardiomyopathies or Alzheimer’s disease [73]. Similarly, lipids are specifically sorted into EVs that are enriched with cholesterol and sphingolipids to give more resistance to chemical and physical changes [69]. Cholesterol acts also in the regulation of EV release [100]. Moreover, ceramide is involved in ILV and MVB production [101].

Recently, special attention has been paid to the nucleic acid content of EVs: Undamaged RNA and miRNAs transported and released into recipient cells according to defined regulatory mechanisms [76]. Increasing evidence demonstrates that RNAs are not passively loaded into EVs because some RNA populationsaremoreabundant than others [69]. It is noteworthy that EVs are particularly rich in 3′UTR (3′ untranslated region) mRNA fragments having multiple sites for miRNA binding [102]. Thereafter, RNA fragments could compete with cellular RNA for binding of miRNAs or proteins contained in the recipient cell and consequently regulate cell stability and translation of proteins. 

However, RNA fragments have a marginal role compared to the miRNAs significant regulation based on post-transcriptional inhibition of genes expression [103]. There is a specific miRNA repertoire exported selectively from EVs, while others are categorically excluded. This demonstrates an active selection mechanism [104,105]. The inclusion of miRNAs into vesicles allows them to circulate in the blood, avoiding degradation by RNase.

## 6. MiRNAs: How Are They Loaded into EVs?

MiRNAs are an emerging class of non-coding RNAs (ncRNAs) ranging between 19 and 24 nucleotides in length that negatively regulate gene expression through complementary interaction with their respective mRNA targets, generally in the 3′UTR [76]. In particular, miRNAs negatively modulate the efficiency of mRNA translation or activate its degradation [106]. It is known that they regulate the expression of approximately 60% of the protein-encoding genes of all mammals [107].

MiRNAs are generated from precursors called primary miRNAs (pri-miRNAs), which are transcribed by RNA polymerase II [108] or RNA polymerase III [109] and undergo a maturational process occurring first in the nucleus (cropping) and then in the cytoplasm (dicing). Pri-miRNAs are cleaved at the 5′ and 3′ ends by Drosha RNase, generating a stem-loop structure of 60–70 nucleotides called pre-miRNA [106]. The pre-miRNA is subsequently translocated into the cytoplasm by Exportin-5 (XPO5) in complex with Ran-GTP. At this point, the RNAse Dicer cleaves off the pre-miRNA loop, forming a miRNA duplex with two nucleotides protruding as overhangs at each 3′ end [110].

Subsequently, one of the two complementary strands of the miRNA duplex is selected to constitute the active miRNA [106]. This miRNA single-strand is associated with the Agonaute2 protein (Ago2) which forms the RNA-induced silencing complex (RISC). The miRNA with the RISC complex is capable of tying the 3′-UTR of the target mRNA [111].

Some studies have reported the presence of Ago2 within EVs [112]; RISC has been shown to associate with MVBsthat produce EVs after fusion with the plasma membrane [113]. In other studies, RISC or Ago2 were not found in EVs [114]. In those cases, heterogeneous nuclear ribonucleoproteins (hnRNP) were thought to play a central role in miRNAs packing into vesicles: Indeed, hnRNPare ubiquitous proteins that are able to recognize and bind some motifs of miRNAs either alone or associated with proteins such as Ago2, Alix, and MEX3C [115]. For example, the nuclear ribonucleoprotein A2B1 (hnRNPA2B1) binds the GGAG motif of miRNAs and controls their loading into EVs [116].

Additional proteins recognize miRNAs motifs, i.e., the synaptotagmin-binding cytoplasmic RNA-interacting protein (SYNCRIP) that binds the GGCU motif [117]. The Alix protein, co-responsible for vesicle biogenesis, is also important in miRNA packing; mutated Alix scales back miRNAs levels but does not affect the amount of EVs [118].

## 7. MiRNAs Found in MSC-EVs: Their Immunoregulatory and Regenerative Properties

The entire pool of miRNAs into MSC-EVs [119] was recently discovered and in silico pathways and networks affected by non-coding RNAs have been identified [120]. However, establishing the contributory effects of a single miRNA is challenging, since many genes and multiples genes can be regulated by more than one miRNA [121]. To accomplish this, researchers have tried different approaches. Some have focused on the most expressed miRNAs in physiological conditions [121,122,123,124,125,126,127,128,129], others have singled out and overexpressed a specific miRNA in EVs [120,127,128,130], used antagomirs [125] or loss of function experiments [129] to evaluate whether EVs are still able to determine a therapeutic effect. In many cases, multiple approaches were adopted simultaneously [122,125,128,129,131,132,133,134]. Overall, it was confirmed that MSC-EV immunoregulatory and regenerative properties can be partially attributed to their nucleic acid content (Table 3).

### 7.1. MiRNAs Involved in Regeneration and Cell Cycle

MiR-196a, miR-27a, and miR-206 present in BMSC (bone marrow stromal cells)-derived EVs are essential to activate osteogenic gene expression (RUNX-2, ALP, OCN, and OPN); furthermore, with the support of a biodegradable hydrogel system, EVs can improve bone regeneration in Sprague Dawley (SD) rats with calvarial defects [122]. Alternatively, miR-133 plays a role in neurite remodeling after stroke in Wistar rats [123].

Several MSC-EV-associated miRNAs play a role in cell cycle regulation and apoptosis. For instance, Ferguson et al. [120] proved that miR-199a targets 22 fundamental genes for cell cycle, such as CABLES1 (Cdk5 and Abl enzyme substrate 1), a cyclin-dependent kinase able to interact with the p53 family of proteins (p53, p63, and p73) [135].

Furthermore, miR-302a inhibits the proliferation of endometrial cancer cells by interfering with the Akt pathway and blocking the overexpression of cyclin D1 [127]; the last is an important proto-oncogene that promotes cell cycle progression from G1 to S phase and acts as transcriptional co-regulator [136]. MiR-155-5p also interferes with the PI3k/Akt signaling pathway in activated B lymphocytes by reducing cell viability. In particular, this miRNA reduces the expression of PAN AKT and phosphorylation of ribosomal protein S6 [137].

Xu et al. [133] have recently shown that exosomal miR-16-5p downregulates ITGA2 (integrin alpha-2) in colorectal cancer (CRC) to stimulate apoptosis and block the proliferation, migration, and invasion of CRC cells.

MiR-126-3p also interact with malignant cells by inhibiting ADAM9 (disintegrin and metalloproteinase domain-containing protein 9) in pancreatic cancer. This type I transmembrane protein is required for tumor progression, highly expressed in metastatic cells [134]. Besides, miR-30 exerts an antiapoptotic effectin renal ischemia injury (IRI). Indeed, this miRNA regulates mitochondrial fission inhibiting Dynamin-related protein 1 (DRP1) on cardiomyocytes [125].

MiR-22 can be effective on myocardial infarction models by targeting methyl CpG binding proteins (Mecp2) [126]. This protein works as an intermediary of DNA methylation and, consequently, is a transcriptional repressor [138] whose inhibition ensures the protection of an ischemic heart [126]. Last but not least, miR-125b-5p suppresses the proapoptotic p53 and BAK1 (Bcl-2 antagonist killer 1) in cardiomyocytes to allow the repair of ischemic damage in vitro and in vivo [131].

### 7.2. Immunoregulatory miRNAs

Some miRNAs exert an immunoregulatory function by acting on specific immunocytes such as miR-15a, miR-15b, and miR-16 that modulate the expression of fractalkine (CX3CL1) in kidneys with ischemia/reperfusion injury. CX3CL1 is a potent chemo-attractant factor for CD86+ macrophages, hence its reduced expression decreases the amount of macrophages migrating to the kidney, alleviating kidney inflammation [124].

MSC-EVs may offer a potential treatment for GVHD; indeed, vesicles contain miR-125a-3p which contribute to the suppression of effector T cell (CD4+ and CD8+) differentiation and maintenance of CD4+, CD25+, Foxp3+ Tregs [121]. Furthermore, miR-146a has therapeutic effects on experimental colitis in rats down-regulating the NF-κB pathway involved in the inflammatory response [130].

EVs also have effect on dendritic cells [119]. Indeed miR-21-5p promotes the degradation of the CCR7 (C–C chemokine receptor type 7) gene, reducing the migration ability of DCs. In addition, miR-142-3p inhibits the expression of the pro-inflammatory cytokine IL-6. Besides, miR-223-3p hinders its maturation, acting on the CD83 gene; miR-126-3p targets Tsc1 (tuberous sclerosis 1), a negative regulator of mTOR kinase which in turn affects cell proliferation and survival [119].

As mentioned before, a single miRNA can exert therapeutic efficacy on multiple targets. This is the case with miR-223, which plays a protective role in both autoimmune hepatitis (by regulating NLRP3 and caspase-1) [128] and in sepsis (by inhibiting Sema3A and Stat3) [129]. NLRP3 (Nacht, LRR, and PYD domains—containing protein3) is expressed in macrophages as a component of the inflammasome. Caspase 1, within the NLRP3 inflammasome complex, activates IL-1β, a mediator of the inflammatory response, whose increased production causes a number of different autoinflammatory syndromes [128]. Sema3A and Stat3 are considered fundamental activators of the sepsis process [129]. Also, miR-126 has many targets and, in particular, inhibits the HMGB1 (High Mobility Group-B1) pathway, with the effect of increasing vascular repair and reducing inflammation in diabetic retinopathy [132].

### 7.3. MiRNAs as Diagnostic Tool

The characterization of miRNAs within EVs and their quantification is also relevant for diagnostic research. Indeed, EVs are secreted by many cell types under physiological conditions and their production considerably increases in pathological conditions [139]. Moreover, they show a different circulating miRNA profile [140]. For example, specific alterations of miRNA expression profiles into EVs have been reported in patients with prostate cancer, breast cancer, glioblastoma, hepatocellular carcinoma, melanoma, pancreatic cancer, and colorectal cancer [141]. Furthermore, the profile of urinary miRNAs have recently been proposed asa diagnostic procedure to identify renal complications of diabetic patients. This urinary miRNA profile could offer a valid alternative to invasive radiological or diagnostic approaches [142]. As regard the diagnostic screening for thyroid carcinoma, Rappa et al. [143] proposed to introduce EVs in “fluid biopsy” as an alternative to invasive tissue biopsy.

## 8. Conclusions

Mesenchymal stem cells have been studied extensively for their potential application in clinical settings. They have regenerative and immunoregulatory properties [13,14,15,21,22,23,24,25,26,27,28,29,30,31,70,71,72] but their use presents a series of limitations, as shown by conflicting results reported in clinical trials. MSC administration in some investigations caused tumors [35,36,37], accumulations of cells or their related by-products in microvasculature [40] or a variable therapeutic effect, influenced by the surrounding micro-environment [13,38,39].

A valid alternative is therefore the use of the MSC secretome which includes cytokines, chemokines, growth factors, anti-inflammatory factors, and EVs [15]. In particular, the latter manages either to produce the same therapeutic effect or is even enhanced in comparison to MSCs [15].

Notably, EVs are simple to preserve, easier to produce [73], and intravenous administration appears to be safer [74,76] and non-toxic [75]. However, further studies are required to develop new techniques to better characterize and standardize vesicles.

MSC and MSC-EVs act through different mechanisms: The therapeutic efficacy of EVs is attributed to their content in nucleic acids, i.e., miRNAs that are selectively loaded by molecular mechanisms involving Ago2 or hnRNPs, SYNCRIP, and Alix [111,112,113,114,115,116,117,118].

Several miRNAs play a role in cell cycle regulation and apoptosis [120,127,133,134,136,137], bone regeneration [122], remodeling neuritis [123] and cardiac repair [126,131]. Other miRNAs already represent a potential treatment for GVHD [121], colitis [130], autoimmune hepatitis [128], sepsis [129], diabetic retinopathy [132], or act directly on peculiar immunocytes [119,124].

Further insights are needed to understand which miRNAs are indispensable and whether theseact simultaneously on multiple genes to achieve a certain effect [121].

In this way, the attention could be shifted from the extracellular vesicle to particular miRNAs. Such miRNAs, when loaded into a system that protects them from the external environment (such as a liposomal envelope), could potentially be used in therapy. Furthermore, these systems could be engineered to be directed towards a specific target cell. Moreover, miRNA investigation offers important perspectives. The identification of particular miRNAs at a given concentration within EVs circulating in the bloodstream would definitely represent an important diagnostic tool [139,140,141,142,143].

## Figures and Tables

**Table 1 ijms-20-04597-t001:** Molecules secreted by mesenchymal stem cells (MSCs) and their functions.

Molecules	Function	In Vitro/In Vivo Models	Ref.
TGF-β	Tregs activation	In vitro: PBMCs from healthy donors In vivo: C57BL/6 and BALB/C mice	[41]
		In vitro: PBMCs from SLE patients In vivo: SLE patients	[42]
	Suppression of DC maturation	In vivo: C57BL/6 and BALB/C mice	[43]
	Suppression of Th17 cell generation	In vivo: MRL/Lpr mice	[44]
mpCCL2	Prevents Th17 chemotaxis and activation	In vivo EAE model	[45]
IL-1Ra	M2-like macrophage polarization	In vivo: Mouse model of arthritis	[46]
	Inhibition of B cell differentiation	In vivo: Mouse model of lung injury	[47]
HGF-1	Preserves renal function	In vivo: Mouse model of obesity-associated kidney injury	[50]
VEGF	Neurogenesis, angiogenesis and reduction of infarct size	In vivo: Mouse model of focal cerebral ischemia	[48]
LIF	Inhibition of Th17 cell differentiation blocking STAT3 phosphorylation	In vivo: EAE model	[51]
	Induction of Foxp3+Tregs proliferation	In vitro: PBMCs from healthy donors	[52]
PGE2	Induces macrophages to produce anti-inflammatory IL-10	In vivo: Mouse model of colitis	[53]
	Promotes macrophages differentiation altering metabolic status and microbicidal capacity	In vitro: THP-1 cell line	[55]
	Promotes hepatocyte proliferation and reduction of apoptosis	In vivo: Mouse model of ALF	[56]
TSG6	Improves tissue repair in peritonitis	In vivo: mouse model of zymosan-induced peritonitis	[57]
	Contributes to inhibiting disease progression	In vivo: EAE model	[58]
	Interferes with leukocyte infiltration and induces corneal regeneration	In vivo: Mouse model of corneal epithelial wound healing in type 1 diabetes	[59]
Gal-1	Inhibits proliferation of CD4+ and CD8+ T cells	In vitro: PBMCs from healthy donors	[60]
Gal-9	Induces death of Th17 and Th1 cells	In vitro: PBMCs from healthy donors	[61]

Abbreviations: Tumor growth factor β (TGF-β), metalloproteinase-processed C–C motif chemokine ligand2 (mpCCL2), interleukin 1 receptor antagonist (IL-1Ra), hepatocyte growth factor-1 (HGF-1), vascular endothelial growth factor (VEGF), leukemia inhibitory factor (LIF), prostaglandin E2 (PGE2), tumor necrosis factor-inducible gene 6 protein (TSG6), galectin-1 (Gal-1), galectin-9 (Gal-9), T helper 17 cells (Th17), signal transducer and activator of transcription 3 (STAT3),Foxp3 (Forkhead box P3), interleukin-10 (IL-10), T helper 1 cells (Th1), systemic lupus erythematosus (SLE), experimental autoimmune encephalomyelitis (EAE), peripheral blood mononuclear cell (PBMCs), acute liver failure (ALF).

**Table 2 ijms-20-04597-t002:** Enzyme activities involved in MSC action.

Enzyme	Activities	In Vitro/In Vivo Models	Ref.
IDO	-It degrades tryptophan into toxic kynurenines - The depletion of tryptophan and toxic components affects the survival of immune cells	In vitro: PBMCs from healthy donors	[62]
NOS	- NOS produces NO: It stops the cell cycle and induces Tcell apoptosis	In vivo: C57BL/6 wild-type mice; C57BL/6 iNOS^−^/^−^ mice	[63]
HO-1	- It degrades heme into biliverdin, iron ions, and CO - Biliverdin compromises infiltration and T cell proliferation - CO inhibits expression of inflammatory cytokines in macrophages	In vivo: Mouse model of cardiac allotransplantation	[64]
		In vivo: Mouse model of ALF	[67]

Abbreviations: indoleamine 2 3-dioxygenase (IDO), nitric oxide synthase (NOS), heme oxygenase-1 (HO-1), NO (nitric oxide), CO (carbon monoxide), peripheral blood mononuclear cell (PBMC), acute liver failure (ALF).

**Table 3 ijms-20-04597-t003:** MiRNAsfound in MSC-derived extracellular vesicles (MSC-EVs), their targets and functions.

MicroRNA ID	Target Proteins	Function	In Vitro/In Vivo Models	Ref.
miR-302a	Inhibits cyclinD1 and Akt signaling pathway	Potential endometrial tumor suppressor	In vitro: Human endometrial cancer cell lines ISK and ECC-1	[127]
miR-199a	Targets 22 genes implicated in cell/death proliferation and cell cycle regulation such as RB1, LKB1, NEUROD1 and CABLES1	Enhances cardiomyocyte proliferation	In silico	[120]
miR-155-5p	PI3K/Akt signaling pathway	Immunomodulatory properties	In vitro: PBMCs from healthy donors; In silico	[137]
miR-125a-3p	//	Suppresses cell proliferation in several cell lines (e.g., effector T cells)	In vitro: PBMCs from healthy donors In vivo: Mouse model of GVHD	[121]
miR-196a miR-27a miR-206	Induce the expression of ALP, OCN, OPN, and Runx2	Regulate osteoblast differentiation	In vitro: Human osteoblasts(hFOB 1.19) In vivo: Sprague Dawley rats	[122]
miR-15a miR-15b miR-16	Modulate CX3CL1 expression	Block infiltration of CD86+ macrophages after ischemia	In vivo: Mouse model of IRI	[124]
miR-22	Methyl-CpG-binding protein 2	Antiapoptotic effect in ischemic cardiomyocytes	In vitro: Neonatal cardiomyocytes In vivo: Myocardial infarction model	[126]
miR-125b-5p	Inhibits p53 and BAK1 in cardiomyocytes	Facilitates ischemic cardiac repair	In vitro: H9C2 cells In vivo: Mouse model of myocardial infarction	[131]
miR-126	Inhibits HMGB1 pathway	Reduces inflammation and improves vascular repair in diabetic retinopathy	In vitro: Human retinal endothelial cells (HREC) In vivo:Diabetic Wistar rats	[132]
miR-16-5p	Down-regulates ITGA2	Blocks proliferation, migration and invasion and enhances apoptosis of colorectal cancer cells	In vitro: CRC cell lines In vivo: BALB/c nude mice	[133]
miR-126-3p	Inhibits ADAM9	Blocks tumor progression and metastasis	In vitro: Pancreatic cancer cell lines In vivo: Xenograft in BALB/c nude mice	[134]
miR-223	Regulates NLRP3 and Caspase-1	Liver protection in autoimmune hepatitis	In vitro: AML12 cells In vivo: Mouse model of autoimmune hepatitis	[128]
	Sema3A and Stat3	Cardioprotection in sepsis	In vivo: Mouse model of CLP	[129]
miR-133b	//	Contributes to neurovascular remodeling after stroke	In vivo: Adult male Wistar rats with stroke	[123]
miR-30	Inhibition of DRP1	Antiapoptotic effect and protection of injured kidney	In vivo: Mouse models of unilateral renal IRI	[125]
miR-146a	Inhibits TRAF6 and IRAK1 and down-regulates NF-κB pathways	Ameliorates experimental colitis	In vivo: Rat model of experimental colitis	[130]
miR-21-5p	CCR7 gene degradation	Attenuate DC maturation and function	In vitro: PBMCs from healthy donors; In silico	[119]
miR-142-3p	Inhibition of IL-6			
miR-223-3p	CD83 gene expression			
miR-126-3p	Negative regulation of Tsc1			

Abbreviations: Retinoblastoma-associated protein (RB1), liver kinase B1(LKB1), neurogenic differentiation 1 (NEUROD1), Cdk5 and Abl enzyme substrate 1(CABLES1), alkaline phosphatase (ALP), osteocalcin (OCN), osteopontin (OPN),runt-related transcript factor-2 (Runx2),fractalkine(CX3CL1), Bcl-2 antagonist killer 1 (BAK1), High Mobility Group-B1 (HMGB1) pathway, integrin alpha-2(ITGA2), disintegrin and metalloproteinase domain-containing protein 9 (ADAM9),Nacht, LRR and PYD domains—containing protein3 (NLRP3), semaphorin-3A (Sema3A),dynamin-related protein 1 (DRP1),TNFreceptor–associated factor 6 (TRAF6), IL-1 receptor-associatedkinase 1 (IRAK1),nuclear factor kappa-light-chain-enhancer of activated B cells(NF-κB) pathways, C–C chemokine receptor type 7 (CCR7), Ishikawa (ISK) cells, human endometrial cancer cells (ECC-1), Graft-versus-Host-Disease (GvHD), human fetal osteoblastic cell line (hFOB), renal ischemia injury (IRI), colorectal cancer cell (CRC), alpha mouse liver 12 (AML12) cell line, cecal ligation and puncture (CLP).

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
