# Peer review of "Insights into the Secretome of Mesenchymal Stem Cells and Its Potential Applications"

_ijms, 2019, doi:10.3390/ijms20184597_

Round 1

Reviewer 1 Report

Content

This manuscript provides a helpful overview of the mesenchymal stem cell secretome and its use as a therapy in a range of in vitro and in vivo models. However, I think this manuscript would be strengthened by focussing entirely on the immunomodulatory properties on the MSC secretome. At present, little is said about the capacity of the secretome to promote endogenous repair mechanisms such as neurogenesis and angiogenesis.

I feel there is an overreliance on referencing other reviews throughout the manuscript. This is particularly apparent in table 1.

Structure

Overall, the manuscript has a clear structure. However, the paragraph in lines 152-164 discussing extracellular vesicles should be moved to section 4. I also recommend leaving spaces between paragraphs to increase the readability.

Language

I feel the manuscript requires thorough proofreading and I have identified some of the typos. The writing is difficult to follow in places and clumsy. It would greatly benefit from being sent to a professional editing service.

Specific Points

Lines 45-47: This sentence is awkward and requires rewording.

Lines 60-62: I do not think it is necessary to list all of the miRNAs here. It makes the sentence difficult to read and these are summarised in table 2.

Line 75: References 1 and 35 are review articles. Primary research articles should be cited here for MSCs maldifferentiating and forming tumours.

Line 82: please clarify what is meant by “inflammatory status”

Line 83: “specific districts”. This is not scientific and vague.

Line 84: Typo. This should be stuck rather than stucked.

Line 143: Table 1 would be a more useful resource for researchers if only primary research papers were cited rather than reviews (e.g. reference 47).

Line 160: EVs are much more challenging to characterise than MSCs are no specific marker have yet been identified. The current recommendations include analysis of surface proteins such as tetraspanins, endosomal proteins e.g. TSG101 and ALIX as well as characterisation of morphology (e.g. TEM, cryo-EM) and size (e.g, dynamic light scattering). Please refer to MISEV 2018 guidelines for further details.

Line 181: I disagree that exosomes are more clearly characterised than MVs. I would be more specific here in the techniques used in characterisation and proteins commonly analysed (e.g. tetraspanins, TSG101). Include a reference to the MISEV 2018 guidelines.

Lines 247-321: I found this section in particular difficult to read. It felt very much like a list.

Line 343:  I feel the self-citation of 4 papers here is inappropriate.

Line 367: typo in abbreviations list. Should be carbon not carbone.

Author Response

Reviewer 1’s comments

This manuscript provides a helpful overview of the mesenchymal stem cell secretome and its use as a therapy in a range of in vitro and in vivo models. However, I think this manuscript would be strengthened by focussing entirely on the immunomodulatory properties on the MSC secretome. At present, little is said about the capacity of the secretome to promote endogenous repair mechanisms such as neurogenesis and angiogenesis.

We acknowledge Reviewer 1 with this comment. We indeed mainly refer throughout the text to immunomodulatory properties of MSC secretome. However, we found many studies about the immunoregolatory effects of a single molecule that consequently has a positive repercussion on tissue repair, but we could not to classify them as a regenerative effect. Following the comment on structure (see below) we took however the opportunity to elaborate another paragraph containing more information for completeness on the overall subject about the regenerative role of EVs and challenging characterization of EVs in toto (lines 163-205)

I feel there is an overreliance on referencing other reviews throughout the manuscript. This is particularly apparent in table 1.

We looked also for primary articles that could replace references of the other reviews: now all the tables have primary articles citations and diseases models.

Structure

Overall, the manuscript has a clear structure. However, the paragraph in lines 152-164 discussing extracellular vesicles should be moved to section 4. I also recommend leaving spaces between paragraphs to increase the readability.

Following Reviewer 1’ s suggestion to move the old lines (152-164) to the next paragraph, we took the opportunity to elaborate another paragraph containing more information about the regenerative role and challenging characterization of EVs in toto (lines 164-205)

Language

I feel the manuscript requires thorough proofreading and I have identified some of the typos. The writing is difficult to follow in places and clumsy. It would greatly benefit from being sent to a professional editing service.

The manuscript underwent linguistic revision and proofreading.

Specific Points

Lines 45-47: This sentence is awkward and requires rewording.

Sentences from line 44 were modified in the revised version

Lines 60-62: I do not think it is necessary to list all of the miRNAs here. It makes the sentence difficult to read and these are summarised in table 2.

Sentence was modified from line 56 of the revised version.

Line 75: References 1 and 35 are review articles. Primary research articles should be cited here for MSCs maldifferentiating and forming tumours.

Review references 1 and 35 are replaced by primary research articles on line 70 of the revised version

Line 82: please clarify what is meant by “inflammatory status”

“Inflammatory status” was clarified on line 77 of the revised manuscript

Line 83: “specific districts”. This is not scientific and vague.

“Some specific districts of the body” was replaced by “microvasculature” on line 80 of the revised manuscript

Line 84: Typo. This should be stuck rather than stucked.

This was corrected on line 81 of the revised version

Line 143: Table 1 would be a more useful resource for researchers if only primary research papers were cited rather than reviews (e.g. reference 47).

The old Table 1 is divided into Table 1 and Table 2 in the revised version and both refer only to primary research articles.

Line 160: EVs are much more challenging to characterise than MSCs are no specific marker have yet been identified. The current recommendations include analysis of surface proteins such as tetraspanins, endosomal proteins e.g. TSG101 and ALIX as well as characterisation of morphology (e.g. TEM, cryo-EM) and size (e.g, dynamic light scattering). Please refer to MISEV 2018 guidelines for further details.

Referring to MISEV 2018 guidelines we deepened on purification and characterization methods for EVs. The old lines 160-161 (“thanks to their structure, MSC-EVs are easier to characterize and standardize than the cells themselves”) were eliminated. See modified paragraph from line 186.

Line 181: I disagree that exosomes are more clearly characterised than MVs.

This sentence was eliminated

 I would be more specific here in the techniques used in characterisation and proteins commonly analysed (e.g. tetraspanins, TSG101). Include a reference to the MISEV 2018 guidelines.

These informations were added in the modified paragraph 4 from line 191 of the revised manuscript.

Lines 247-321: I found this section in particular difficult to read. It felt very much like a list.

Lines 298-381 of the revised version (the old lines 247-321): we tried to make reading more comfortable by  resing structure and introducing subparagraphs were single miRNAs were discussed according to their specific effects or use.

Line 343:  I feel the self-citation of 4 papers here is inappropriate.

We considered the most appropriate citation on line 405 of the revised version.

Line 367: typo in abbreviations list. Should be carbon not carbone.

This was corrected on the revised Abbreviations list starting on line 431.

Reviewer 1’s comments

This manuscript provides a helpful overview of the mesenchymal stem cell secretome and its use as a therapy in a range of in vitro and in vivo models. However, I think this manuscript would be strengthened by focussing entirely on the immunomodulatory properties on the MSC secretome. At present, little is said about the capacity of the secretome to promote endogenous repair mechanisms such as neurogenesis and angiogenesis.

We acknowledge Reviewer 1 with this comment. We indeed mainly refer throughout the text to immunomodulatory properties of MSC secretome. However, we found many studies about the immunoregolatory effects of a single molecule that consequently has a positive repercussion on tissue repair, but we could not to classify them as a regenerative effect. Following the comment on structure (see below) we took however the opportunity to elaborate another paragraph containing more information for completeness on the overall subject about the regenerative role of EVs and challenging characterization of EVs in toto (lines 163-205)

I feel there is an overreliance on referencing other reviews throughout the manuscript. This is particularly apparent in table 1.

We looked also for primary articles that could replace references of the other reviews: now all the tables have primary articles citations and diseases models.

Structure

Overall, the manuscript has a clear structure. However, the paragraph in lines 152-164 discussing extracellular vesicles should be moved to section 4. I also recommend leaving spaces between paragraphs to increase the readability.

Following Reviewer 1’ s suggestion to move the old lines (152-164) to the next paragraph, we took the opportunity to elaborate another paragraph containing more information about the regenerative role and challenging characterization of EVs in toto (lines 164-205)

Language

I feel the manuscript requires thorough proofreading and I have identified some of the typos. The writing is difficult to follow in places and clumsy. It would greatly benefit from being sent to a professional editing service.

The manuscript underwent linguistic revision and proofreading.

Specific Points

Lines 45-47: This sentence is awkward and requires rewording.

Sentences from line 44 were modified in the revised version

Lines 60-62: I do not think it is necessary to list all of the miRNAs here. It makes the sentence difficult to read and these are summarised in table 2.

Sentence was modified from line 56 of the revised version.

Line 75: References 1 and 35 are review articles. Primary research articles should be cited here for MSCs maldifferentiating and forming tumours.

Review references 1 and 35 are replaced by primary research articles on line 70 of the revised version

Line 82: please clarify what is meant by “inflammatory status”

“Inflammatory status” was clarified on line 77 of the revised manuscript

Line 83: “specific districts”. This is not scientific and vague.

“Some specific districts of the body” was replaced by “microvasculature” on line 80 of the revised manuscript

Line 84: Typo. This should be stuck rather than stucked.

This was corrected on line 81 of the revised version

Line 143: Table 1 would be a more useful resource for researchers if only primary research papers were cited rather than reviews (e.g. reference 47).

The old Table 1 is divided into Table 1 and Table 2 in the revised version and both refer only to primary research articles.

Line 160: EVs are much more challenging to characterise than MSCs are no specific marker have yet been identified. The current recommendations include analysis of surface proteins such as tetraspanins, endosomal proteins e.g. TSG101 and ALIX as well as characterisation of morphology (e.g. TEM, cryo-EM) and size (e.g, dynamic light scattering). Please refer to MISEV 2018 guidelines for further details.

Referring to MISEV 2018 guidelines we deepened on purification and characterization methods for EVs. The old lines 160-161 (“thanks to their structure, MSC-EVs are easier to characterize and standardize than the cells themselves”) were eliminated. See modified paragraph from line 186.

Line 181: I disagree that exosomes are more clearly characterised than MVs.

This sentence was eliminated

 I would be more specific here in the techniques used in characterisation and proteins commonly analysed (e.g. tetraspanins, TSG101). Include a reference to the MISEV 2018 guidelines.

These informations were added in the modified paragraph 4 from line 191 of the revised manuscript.

Lines 247-321: I found this section in particular difficult to read. It felt very much like a list.

Lines 298-381 of the revised version (the old lines 247-321): we tried to make reading more comfortable by  resing structure and introducing subparagraphs were single miRNAs were discussed according to their specific effects or use.

Line 343:  I feel the self-citation of 4 papers here is inappropriate.

We considered the most appropriate citation on line 405 of the revised version.

Line 367: typo in abbreviations list. Should be carbon not carbone.

This was corrected on the revised Abbreviations list starting on line 431.

Reviewer 2 Report

1.This is a review article. Authors should make a figure to summarize the content and potential effect of the secretome of MSC.

2. In table 1 and 2, the column should show which disease models?

3. The subtitles of manuscript should be more divided, such as subtitle 3. the secretome of MSCs change to 3A. molecules in the secretome of MSCs and 3B. enzyme activity in the secretome of MSCs.....

4. Abstract does not include most important information of manuscript which need to be reorganized.

5. English need further editing.

Author Response

1.This is a review article. Authors should make a figure to summarize the content and potential effect of the secretome of MSC.

Figure 1 was inserted in paragraph 4 that could work also as Graphical Abstract

In table 1 and 2, the column should show which disease models?

Disease models were inserted in the revised Tables

The subtitles of manuscript should be more divided, such as subtitle 3. the secretome of MSCs change to 3A. molecules in the secretome of MSCs and 3B. enzyme activity in the secretome of MSCs.....

We followed this suggestion in numbering paragraphs within the revised version

Abstract does not include most important information of manuscript which need to be reorganized.

An improved version of the Abstract was inserted following the Reviewer 2’s suggestion

English need further editing

The manuscript underwent English revision.

Round 2

Reviewer 1 Report

I would like to thank the authors for addressing my comments and putting a considerable amount of work into the manuscript. I believe it has been greatly improved.

Line 63: EV represent rather than represents

Line 138: Growth factor including not growth factors as

Line 286: this sentence is clumsy and does not make sense

Line 456: evidence not evidences

Line 723: typo “toto”

I think Figure 1 is not very detailed and does not enhance the reader’s understanding of the review. I would recommend removing it.

Author Response

I would like to thank the authors for addressing my comments and putting a considerable amount of work into the manuscript. I believe it has been greatly improved.

Line 63: EV represent rather than represents

Line 138: Growth factor including not growth factors as

Line 286: this sentence is clumsy and does not make sense

We hope to have identified correctly this sentence since lines number identification was difficult

Line 456: evidence not evidences

Line 723: typo “toto”

Suggested corrections were inserted in the revised version of the manuscript

I think Figure 1 is not very detailed and does not enhance the reader’s understanding of the review. I would recommend removing it.

Figure 1 was deleted as recommended by this Reviewer

Reviewer 2 Report

The quality and content of figure 1 were inadequate. The miRNA should be pointed from EVs. In addition, word size and graphs should be more delicate.

Author Response

The quality and content of figure 1 were inadequate. The miRNA should be pointed from EVs. In addition, word size and graphs should be more delicate.

Figure 1 was deteled (see also reply to Reviewer 1). We believe that content was clearly detailed in added Tables therefore would not be exhaustive.